# PROVCREATOR: SYNTHESIZING GRAPHS WITH TEXT ATTRIBUTES

## ABSTRACT

In cybersecurity, system provenance graphs are a key primitive to support intrusion detection and program identification tasks. Recent movement towards using data-hungry graph learning models for security-critical applications has exposed significant limitations in existing provenance datasets. Imbalanced representation of programs induces bias and performance degradation in downstream models. Further, these models rely on rich numeric and textual node attributes to accurately encode program behaviors, limiting the ability of existing data augmentation techniques to address data imbalance in provenance graphs.

We present PROVCREATOR, a novel graph synthesis framework designed for feature-rich system provenance graphs. PROVCREATOR learns the joint distribution of node attributes and graph structures conditioned on program class labels, enabling targeted generation of realistic system provenance graphs to supplement underrepresented programs. Our evaluation shows that PROVCREATOR produces provenance graphs with higher structural fidelity, attribute fidelity, and downstream utility compared to those of previous graph synthesis methods.

## 1 INTRODUCTION

System provenance (Inam et al., 2023) has gained prominence in recent years as a powerful tool to counter advanced cyber attack campaigns. The seminal work of King et al. (King & Chen, 2003) established system provenance as a vital field for comprehensive system-wide surveillance and protection. In system provenance analysis, causal graphs are extracted from surveillance logs by connecting resources through syscall-level access events. As outlined in Table 1, in provenance graphs, nodes represent system entities (*e.g.,* processes, files, and network sockets), while edges denote the interactions between these entities, including `CREATE` a process, `READ` or `WRITE` a file, or `SEND` or `RECEIVE` data to or from a network socket. These nodes have security-relevant attributes such as filenames, executable names, and IP addresses. Provenance graphs are consumed by Machine Learning (ML) models to perform security critical tasks, such as intrusion detection (Rehman et al., 2024) and program classification (Mukherjee et al., 2023).

Table 1: Provenance dataset schema.

| Nodes | Edges |
|---|---|
| Process | ProcessEvent (Process → `CREATE` → Process) |
| File | FileEvent (Process → `READ`, `WRITE`, `EXECUTE` → File) |
| IPChannel | IPChannelEvent (Process → `SEND`, `RECEIVE` → IPChannel) |

However, the effectiveness of ML models is hampered by the inherent data imbalance in system provenance datasets, as shown in Figure 1. These datasets often lack sufficient representation of program behaviors, particularly when it comes to less common or underrepresented runtime configurations such as `WaaSMedicSvc` (label D) and `DoSvc` (label E) of `svchost.exe` and `Write-Host` (label C), `$Infs = Get-Item` (label D), and `Get-AppxPackage` (label E) of `powershell.exe`. Such imbalance introduces systematic bias in the training process, leading to poor generalizability of security models. A balanced system provenance dataset is crucial for ensuring reliable intrusion detection. An ideal dataset should include equal representation of all relevant programs, as well as their various runtime configurations, such as command-line arguments. The imbalance found in

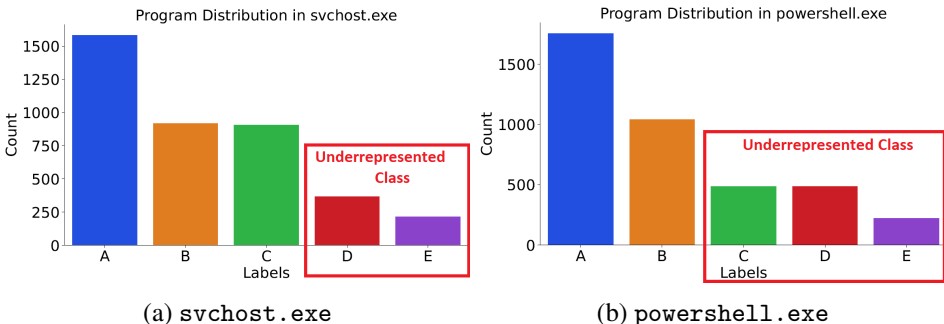

(a) `svchost.exe`  (b) `powershell.exe`

Figure 1: Provenance dataset of `svchost.exe` and `powershell.exe` showcase that certain programs are underrepresented in the dataset causing dataset imbalance. The underrepresented program being labelled D and E for `svchost.exe`, and C, D, and E for `powershell.exe`. For detailed descriptions of these labels, refer to §A.1.

real-world datasets means that ML models trained on these datasets may fail to generalize to unseen environments (Zhou et al., 2023). This could reduce their effectiveness in practical deployments (Alsaheel et al., 2021) as models may incorrectly predict the behavior of underrepresented programs, leading to false positives in ML-based intrusion detection systems.

In this work, we introduce PROVCREATOR, a novel graph synthesis framework designed to address the under-representation of program behaviors in system provenance datasets by synthesizing realistic provenance graphs. These synthetic graphs maintain both structural and attribute-wise similarity to the original data. To the best of our knowledge, this is the first approach within the provenance domain to simultaneously generate both graph structure and conditionally accurate node attributes. Outside the provenance domain, previous works such as Jo et al. (2022) have focused on generating graph structures with categorical attributes, but have not tackled the challenge of learning textual node attributes conditioned on graph structure — an inherently complex task, as node attributes are influenced by both local neighborhood and distant graph nodes.

Beyond just generating a uniform and fixed set of text attributes for all nodes, PROVCREATOR offers the flexibility to associate nodes with arbitrary sets of text attributes, each with distinct semantics that can be incorporated and modified with no architecture changes. The ability to synthesize multiple text attributes for each node with distinct semantics has applications across many domains where graphs contain semantically rich attributes. While our work only considers static attribute sets for each node type, PROVCREATOR's architecture naturally allows for dynamically determined node attribute sets using flexible attribute indicators.

By jointly generating the graph structure and node attributes, PROVCREATOR is able to achieve better structural and attribute-wise fidelity with the original training distribution compared to prior works, and is able to better support downstream model performance for both classification and intrusion detection. In summary, this work makes the following contributions:

- **Data Imbalance Mitigation.** We introduce PROVCREATOR, a novel graph synthesis framework that generates balanced provenance data to address the systematic bias caused by underrepresented program behaviors in existing datasets.
- **Improved Model Performance.** We show that ML models trained with PROVCREATOR-augmented data perform significantly better when compared against those trained on data generated by prior graph synthesis methods.
- **Flexible Text Attribute Generation.** Our PROVCREATOR attribute generation model is able to learn diverse sets of text attributes and accurately generate those attributes using the structure of the graph.

While the proposed research focuses on the synthetic generation of system provenance graph datasets, we believe that the PROVCREATOR approach can be applied to other security graph datasets or highly complex, heterogeneous graphs with textual and numerical attributes. To benefit the research community, we plan to publicly release our augmentation framework, as well as the datasets generated in this study, after applying privacy-preserving measures to ensure data confidentiality.

## 2 RELATED WORK

**Synthetic Data Generation.** Synthetic data generation (Yang et al., 2022) involves creating artificial data that closely resembles real data in its statistical properties and complexity without collecting additional real data. These approaches aim to address challenges presented by imbalanced datasets by supplementing underrepresented classes with synthetic examples. Recent advancements in data analysis techniques and data processing have led to numerous research proposals for data synthesis across various domains (Cubuk et al., 2019; Al Olaimat et al., 2020; Cordero et al., 2021; Feng et al.; Yan et al., 2022; Zhao et al.; 2022). Current synthetic graph generation methods consider graph structure generation and attribute generation separately. We argue that they should be considered together because neighbourhood structures are often stastistically codependent with node attributes. Therefore, PROVCREATOR conditionally generates node attributes based on node neighborhood embeddings. To confer structural understanding onto the attribute generation process, we jointly train a graph encoder with a transformer-based attribute generation model. Combined with incremental improvements in graph structure generation, this formulation yields state-of-the-art synthetic graphs with rich text attributes.

Although synthetic graph generation research has attracted significant attention across various research domains, security datasets, particularly system provenance graphs, have seen limited progress. Synthetic data is valuable in the system provenance domain because real `syscall` traces are costly to collect and difficult to share due to privacy concerns, but are required in large quantities by ML models. Creech & Hu (2013) and Haider et al. (2016) proposed initial prototype implementations for synthesizing attack behaviors by generating sequences of system calls, but these implementations lacked consideration of causal dependencies between system resources (*e.g.,* processes, files, and network sockets). The operating system (OS) used in Creech & Hu (2013) is decade old Ubuntu 11.04 which has reached end of life support and it contains no Windows traces. Furthermore, Creech & Hu (2013) rejected traces over 3 kB thus impacting the completeness of the traces collected. By directly generating provenance graphs, PROVCREATOR synthesizes data in an OS-agnostic way with structural consideration of causal dependencies, and can generate large graphs by combining related subgraphs. While synthetic data generation has been explored in other security research domains (Cordero et al., 2021), we are unaware of any previous research focusing on synthetic data generation approaches specifically for system provenance graph datasets.

**System Provenance.** System provenance tracks fine-grained system data (`syscall` events), from large enterprise and industrial systems. It traces bidirectional information flow and control dependencies starting from a Point-Of-Interest (POI) event, enabling forensic analysis and advanced security defenses (King & Chen, 2003; Liu et al., 2018; Hassan et al., 2019; Wang et al., 2020). By examining system-call logs (aud, 2015; etw, 2019; Cantrill, 2005), system provenance graphs capture relationships (*i.e.,* READ, WRITE, CREATE and EXECUTE) among major resources (*i.e.,* processes, files and network sockets). Nodes, representing system resources, are annotated with attributes such as executable names, filenames, and IP addresses, making these graphs invaluable tools for forensic analysis to discover points of entry, track infection propagation, and assess the scale of damage. Formally, a provenance graph is a connected set of timestamped edges $e = (u, v, r)$, where $u, v \in \{processes \cup files \cup sockets\}$ and $v$ is causally dependent on $u$ (*e.g.,* a file $u$ is written by a process $v$), and $r$ is the relationship between the nodes.

**Provenance-based ML Research.** Initially proposed to automate forensic investigations, system provenance has become a vital foundation for ML-based security detectors (Wang et al., 2020; Han et al., 2020; 2021; Jia et al., 2024; Cheng et al., 2024; Mukherjee et al., 2024; Rehman et al., 2024), propelled by rapid advances in data analysis techniques. Security researchers have expanded the scope of system provenance studies to construct ML models that counter attacks where adversaries craft unique, unexposed attack vectors (Mukherjee et al., 2023) that can only be identified during runtime. While the full provenance graph even for one host machine can be intractably large, comprising millions of edges, analysis can be performed on tractable subgraphs extracted by tracing causal dependencies to and from a POI event. Depending on the objective of the analysis task, various query conditions can determine effective POI events, with the system provenance graph serving as the primary target for analysis.

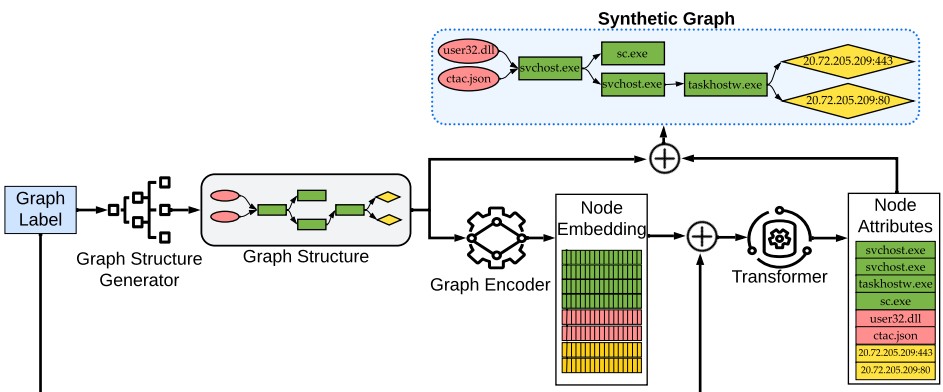

Figure 2: PROVCREATOR's synthetic graph generation pipeline. Given a graph class label, PROVCREATOR first generates the graph structure, then constructs node attributes based on the class label and the generated structure. The result is a synthetic graph with node attributes that are contextually consistent.

## 3  GRAPH SYNTHESIS METHODOLOGY

PROVCREATOR focuses on two key aspects of graph generation: the generation of the graph structure and the prediction of node attributes within the generated graph. Previous research has primarily focused on individual aspects without attempting to combine graph structure and graph attribute generation tailored for cybersecurity applications. Our work is the first to integrate both approaches within the cybersecurity domain. System provenance graphs are directed heterogeneous graphs, where different node types are associated with varying attributes. For instance, processes have executable names, files have filenames, and sockets have IP addresses and ports.

Figure 2 depicts PROVCREATOR, which trains on system provenance graphs with known labels. In §3.1, PROVCREATOR extends Graph Diffusion Score-based Sampling (GDSS) (Jo et al., 2022) to learn the structure of graphs. It then encodes the graph structure as a set of node embeddings. In §3.2, these node embeddings are then refined through a transformer model, which is trained to reconstruct node attributes based on the learned embeddings. This enables the model to accurately predict the complex attributes associated with each node. Finally, the two components (graph structure and node attributes) are integrated in §3.3 by applying the generated node attributes to the generated graph structure. This process ensures both the accurate generation of graph structures and the effective prediction of node attributes, which are essential for semantically rich heterogeneous graphs.

### 3.1  STRUCTURE GENERATION

Consider a heterogeneous graph $G(V, E)$, where each node $v \in V$ and edge $e \in E$ belongs to certain pre-defined types. Each node and edge can have an associated list of attributes, such as executable path, filename, IP address, *etc.* Assume that all real-world provenance graphs belong to a distribution $p(G|c)$, where $c$ is the context of the graph, such as the command line of the POI process. The goal of graph structure generation is to learn a deep learning model $\mathcal{F}_{\theta_F}(c, z) \to G$ to generate graphs that follow the distribution $p$, given the condition $c$ and random vector $z$.

For the design of the model $\mathcal{F}$, many different approaches have been developed in the literature of general graph generation. Models such as Kipf & Welling (2016), Simonovsky & Komodakis (2018) generate the whole graph at once. Work such as You et al. (2018), Wang et al. (2017) considers step-by-step generation where the model iteratively adds nodes and edges to the graph.

In our work, we consider a more recent diffusion based approach based on Jo et al. (2022). To model the problem, we relax the adjacency matrix representation $A$ of the graph to real numbers $A \in \mathbb{R}^{|V| \times |V|}$. A homogeneous graph with node features $F$ can then be represented as $G = (X, A)$ where $X \in \mathbb{R}^{|V| \times |F|}$. For heterogeneous graph, node types can be treated as part of the node features and edge types can be generated by extending the adjacency matrix with an extra dimension for edge features $A \in \mathbb{R}^{|V| \times |V| \times |E|}$. The design of $E$ can be different depending on the dataset, for

our provenance dataset, we use one-hot embedding of the categorical actions described in Table 1. The $T$ step diffusion process can be represented as the trajectory of $\{G_t = (X_t, A_t)\}_{t \in [0,T]}$, where $G_0$ is the original graph and $G_t$ follows a prior distribution such as Gaussian distribution. The model $\mathcal{F}$ learns the reverse of the diffusion process and generates the graph from $G_t$ which can be randomly sampled with random vector $z$.

The original GDSS paper (Jo et al., 2022) uses a GCN as the backbone and it does not consider conditional generation. We improved the backbone model by using a graph transformer (Dwivedi & Bresson, 2020) and following the original stable diffusion (Rombach et al., 2021) paper's approach to add the conditioning mechanism for the graph transformer. The condition is set based on the label of the process represented in the graph; details about the labels can be found in §A.1.

## 3.2 Attribute Generation

Most graph generation models only consider simple categorical attributes on nodes and edges. GDSS (Jo et al., 2022) claims they are the first work that proposes a diffusion process for generating a whole graph consisting of nodes and edges with attributes. They generate molecular graphs where the node attributes are the elements and the edge attributes are the types of chemical bonds between the atoms. In our dataset, in addition to the node types and edge types shown in Table 1, we have additional string attributes associated with the nodes, such as executable paths and filenames. Generating these attributes is challenging, because: *(1)* string attributes have variable length and can be hundreds of characters long, which makes representing them in a graph neural network difficult; *(2)* different node types have distinct and nonuniform sets of associated attributes, so the attribute generation must have special considerations for each node type. Earlier works such as Koncel-Kedziorski et al. (2019) consider graph-level text extraction from knowledge graphs. In contrast, we consider node-level text generation purely based on graph structure and a class label. That is, our goal is to learn an attribute generation model $\mathcal{T}_{\theta_T}(h_v, c, a) \rightarrow s$, that produces a text attribute $s$ given a node embedding $h_v$, a context vector $c$, and an attribute indicator $a$.

Given a generated structure-only heterogeneous graph with node types and edge types, a Graph Neural Network (GNN) encoder $\mathcal{E}_{\theta_E}(G)$ is used to extract the node embedding $h_v$ for each node $v \in G$. Then for each node we input the node embedding to the attribute generation model $\mathcal{T}_{\theta_T}(h_v, c, a)$ with the context vector $c$ and attribute indicator $a$. The attribute indicator is a text label associated with a type of attribute to generate. In our work, each node type has a fixed list of associated attribute indicators (*e.g.,* network sockets require both IP addresses and port numbers). Specifically, we use a transformer decoder for $\mathcal{T}_{\theta_T}$. Since the transformer decoder is an auto-regressive model, we put $h_v, c, a$ as the initial (embedding of) input tokens. Then a <bos> token is added to indicate the beginning of the output. During training, we apply a mask similar to what one would apply in a translation task, where the node embedding input is unmasked, and the target tokens after <bos> are masked with a lower triangular matrix so at each token location the model can get attention from both the embedding vector and the previously generated token. The attention masking strategy is commonly seen in seq2seq models, and can also be found in image-to-text models such as Wang et al. (2022). The introduction of the attribute indicator $a$ enables the model to generate different attributes for different types of nodes with the flexibility to extend the list of attributes without architecture changes.

## 3.3 Structure and Attribute Integration

To generate a graph $G$, we first generate its structure $G = \mathcal{F}_{\theta_F}(c, z)$. Then we obtain the structural node representation with the graph encoder $\{h_v\}_{v \in G} = \mathcal{E}_{\theta_E}(G)$. For each node embedding $h_v$ in $\mathcal{E}_{\theta_E}(G)$, and for each attribute indicator $a$ that needs to be generated, we use the attribute generation model $\mathcal{T}_{\theta_T}$ to calculate $\mathcal{T}_{\theta_T}(h_v, c, a)$ yielding each attribute. The output graph is assembled by attaching the generated attributes to the corresponding nodes. The graph encoder $\mathcal{E}_{\theta_E}$ is trained together with the attribute generation model $\mathcal{T}_{\theta_T}$, and the generation of different attributes are trained jointly. The algorithm is detailed in Algorithm 1.

---

**Algorithm 1** Structure and Attribute Integration Training

---

**Input:** Graph dataset $\mathcal{D}$, learning rate for encoder model $\eta_E$, learning rate for attribute generation model $\eta_T$, and number of epochs $N_{epoch}$
**Output:** Trained graph encoder model $\mathcal{E}_{\theta_E}$ and attribute generation model $\mathcal{T}_{\theta_T}$
Initialize graph encoder model parameters $\theta_E$
Initialize attribute generation model parameters $\theta_T$
**for** epoch $= 1$ to $N_{epochs}$ **do**
 $\delta_{\theta_E} \leftarrow 0$
 **for** each graph $G$ and context vector $c$ in $\mathcal{D}$ **do**
  $\{h_v\}_{v \in G} \leftarrow \mathcal{E}_{\theta_E}(G)$
  $B \leftarrow \{\}$             ▷ Set to accumulate batch training samples
  **for** each $v \in G$ **do**     ▷ each $v$ contains a collection of indicator-attribute $(a, s)$ pairs
   **for** each node attribute indicator $a$ in $v$ and corresponding node attribute $s$ **do**
    Add $(h_v, a, s)$ into $B$
   **end for**
  **end for**
  **for** each batch of node attributes $(h_{\text{batch}}, a_{\text{batch}}, s_{\text{batch}})$ in $batch(B)$ **do**
   $s' \leftarrow \mathcal{T}_{\theta_T}(h_{\text{batch}}, c, a_{\text{batch}})$
   $l = \mathcal{L}_{\text{MLM}}(s', s_{batch})$        ▷ $\mathcal{L}_{\text{MLM}}$ is masked language loss
   $\theta_T \leftarrow \theta_T - \eta_T \nabla_{\theta_T} l$          ▷ Update $\mathcal{T}_{\theta_T}$
   $\delta_{\theta_E} \leftarrow \delta_{\theta_E} + \nabla_{\theta_E} l$     ▷ Accumulate the gradient for graph encoder $\mathcal{E}_{\theta_E}$
  **end for**
 **end for**
 $\theta_E \leftarrow \theta_E - \eta_E \delta_{\theta_E}$             ▷ Update $\mathcal{E}_{\theta_E}$
**end for**
**Return:** Trained model $\mathcal{E}_{\theta_E}, \mathcal{T}_{\theta_T}$

---

## 4 EVALUATION

To comprehensively evaluate PROVCREATOR, we first demonstrate direct improvements over Jo et al. (2022) in graph structure generation (§4.1), then demonstrate competent results against strong baselines in textual attribute generation (§4.2) and overall graph composition (§4.3). Finally, we demonstrate improvements over Jo et al. (2022) in supporting downstream supervised and unsupervised graph learning tasks (§4.4).

**Evaluation Protocol.** We trained PROVCREATOR and Jo et al. (2022) to generate provenance graphs for svchost.exe and powershell.exe in a Windows environment, then used them to supplement underrepresented sub-programs in our datasets. We chose these programs because they are popular targets for impersonation by advanced cyber threats (Barr-Smith et al., 2021), and their behavior is largely determined by their command-line arguments, which provides clear class labels.

For graph similarity measures (§4.1, §4.2, and §4.3), we generated 1,000 graphs for each of svchost.exe and powershell.exe with PROVCREATOR and with Jo et al. (2022) using the same training set, then measured the distributional distance of the synthetic graphs to the training set with established metrics. We additionally propose a node attribute similarity metric with domain-specfic considerations, BLEU+, which is described in §A.2. For metrics that rely on textual attributes (§4.2 and §4.3), we augment the synthetic graphs from Jo et al. (2022) with appropriate randomly sampled node attributes from the training set to form a strong baseline.

For downstream tasks (§4.4), we refer to respected works from the cybersecurity domain to guide our supervised program classification task (Barr-Smith et al., 2021) and unsupervised malware detection task (Rehman et al., 2024). In each of these tasks, we train the downstream GNN models on each of: *(1)* the original training data; *(2)* the original training data supplemented with graphs generated by Jo et al. (2022); and *(3)* the original training data supplemented with graphs generated by PROVCREATOR. For each downstream task, the relevant scoring metrics are reported for each model configuration.

Table 2: Maximum mean discrepancy (MMD) distances of different graph metrics between real and synthetic datasets generated with GDSS (Jo et al., 2022) and PROVCREATOR where lower number is better.

| | Degree ↓ | Clustering ↓ | Bet. Cen. ↓ | Cls. Cen. ↓ | Katz Cen. ↓ | Spectral ↓ |
|---|---|---|---|---|---|---|
| | | | svchost.exe(win) | | | |
| GDSS | 0.038 | 0.022 | 0.008 | 0.228 | 0.209 | 0.115 |
| PROVCREATOR | **0.030** (-0.008) | **0.002** (-0.020) | **0.007** (-0.001) | **0.175** (-0.053) | **0.068** (-0.141) | **0.060** (-0.055) |
| | | | powershell.exe(win) | | | |
| GDSS | 0.013 | 0.267 | 0.010 | 0.164 | 0.064 | 0.066 |
| PROVCREATOR | **0.003** (-0.010) | **0.048** (-0.219) | **0.007** (-0.003) | **0.094** (-0.070) | **0.044** (-0.020) | **0.021** (-0.045) |

## 4.1 STRUCTURE FIDELITY

Table 2 measures structural fidelity for each of several graph structural stastistics between the real and synthetic datasets, mirroring the evaluation of Jo et al. (2022). The maximum mean discrepancy (MMD) distances were calculated between our test datasets and synthetic datasets generated from PROVCREATOR and GDSS models trained on our `svchost.exe` and `powershell.exe` datasets. In both datasets, we observed a smaller MMD distance for PROVCREATOR than GDSS in all metrics. This indicates an improvement in structure generation quality over GDSS due to better disambiguation between graph categories through the context vector $c$, along with incremental improvements to the base model architecture.

Table 3: Process, file, and IP channel attribute similarity comparison between real and synthetic datasets. The similarity of process and file attributes is evaluated with BLEU (Papineni et al., 2002) and BLEU+. IP channel attribute accuracy is measured based on the network segment of the IP address and the value of the port.

| | Process BLEU ↑ | Process BLEU+ ↑ | File BLEU ↑ | File BLEU+ ↑ | IPChannel IP Accuracy ↑ | IPChannel Port Accuracy ↑ |
|---|---|---|---|---|---|---|
| | | | svchost.exe(win) | | | |
| Baseline | 0.555 | 0.989 | 0.919 | 0.933 | 0.135 | 0.681 |
| PROVCREATOR | **0.520** (-0.035) | **0.995** (+0.060) | **0.995** (+0.076) | **0.996** (+0.063) | **0.258** (+0.122) | **0.258** (-0.423) |
| | | | powershell.exe(win) | | | |
| Baseline | 0.792 | 0.841 | 0.930 | 0.907 | 0.395 | 0.998 |
| PROVCREATOR | **0.842** (+0.050) | **0.922** (+0.081) | **0.990** (+0.060) | **0.982** (+0.075) | **0.838** (+0.443) | **0.839** (-0.159) |

## 4.2 ATTRIBUTE FIDELITY

Provenance graph attributes are complex and domain-specific, making their evaluation challenging. This section focuses on evaluating individual attributes. For processes and files, we evaluated the generated filenames and executable names with BLEU and BLEU+ within each respective node type. To reduce semantically unimportant noise in IP addresses, we only consider 6 coarse-grained network segments: private, multicast, global, loopback, link-local, and other. A generated IP address is correct when it is valid and belongs to the same segments as the ground truth IP address. Similarly, a generated port is correct either when it exactly matches the ground truth port, or when the generated port and the ground truth port are both in the range [10,000, 65,535]. The reported accuracies for IP addresses and ports are then simply the proportion of correct generations.

Because synthetically generated graph structures have no ground truth attributes, the experiments in this section are conducted by generating attributes for real graph structures. The results are shown in Table 3. The simple baseline method (randomly sampling node attributes from the correct node type in the training set) is strong due to how the metrics are defined. For example, if the sampled text matches that of any node in the graph, BLEU and BLEU+ will be 1.0, making the baseline model the upper bound for models that ignore graph structure. To outperform this baseline, not only do the generated text attributes need to be accurate, they also need to be correctly associated with each specific graph. With this in mind, our proposed method achieves significant improvements, indicating that not only are the generated attributes realistic, they also correctly consider the graph structural context. Further, we observed that BLEU and BLEU+ can come to opposite conclusions on the same data, indicating that domain-specific evaluation is necessary. The port numbers are generated as text tokens and converted to integers, which is generally a bad practice, and not surprisingly, resulted in poor performance. We leave effective numeric attribute generation for future work.

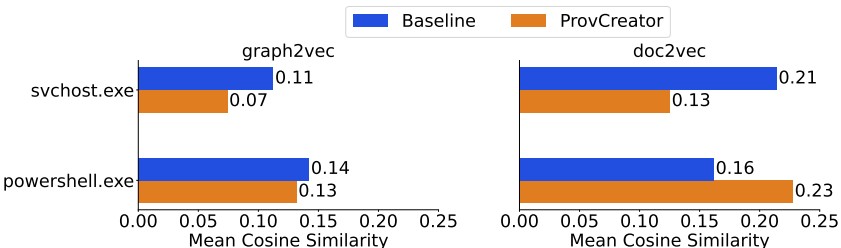

Figure 3: Embedding cosine similarity comparison between real and synthetic `svchost.exe` and `powershell.exe` datasets using graph2vec (Narayanan et al., 2017) and doc2vec (Le & Mikolov, 2014a).

### 4.3 Embedding Fidelity

We evaluate the overall similarity of the synthetic graphs to the original training set by embedding the graphs into vectors, then measuring the average cosine similarity of those graph embeddings to the embeddings of the training set. In Figure 3, we compare PROVCREATOR's synthetic graphs to those of Jo et al. (2022), augmented with appropriately randomly sampled node attributes from the training set. For completeness, we use two embedding schemes: doc2vec (Le & Mikolov, 2014b), which creates a text document from a graph through a series of random walks, and graph2vec (Narayanan et al., 2017), which extends doc2vec with hierarchical graph structure representations. Figure 3 shows that PROVCREATOR is able to competently compare even against synthetic graphs that have node attributes directly sampled from the training set.

### 4.4 Downstream Application

To demonstrate the practical utility of PROVCREATOR, we consider two security-relevant downstream tasks: program classification and malware detection. Figure 4 and Table 4 show classification and malware detection results, respectively.

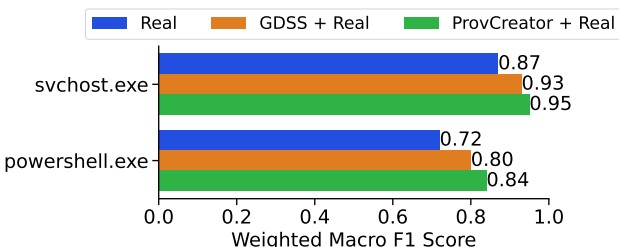

Figure 4: Weighted macro F1 scores for subprogram classification in `svchost.exe` and `powershell.exe`, showing improved performance from adding PROVCREATOR's synthetic provenance graphs.

**Program Classification.** Program classification is a supervised learning task in which the model aims to identify the program being executed by a host process. For example, given a provenance graph centered around an `svchost.exe` process, identify which command was run based on the system resources that appear in the graph. The full list of class labels considered in this task is provided in §A.1.

In Figure 5, we see that although the baseline performance is quite strong, there are still a couple classes that are confused. While existing methods (Jo et al., 2022) are able to provide some improvement, most of the confusion in underrepresented program classes remains. PROVCREATOR significantly reduces confusion in those underrepresented programs by guiding its graph synthesis to focus on supplementing those minority classes. These observations are consistent across both `svchost.exe` and `powershell.exe` classification tasks, demonstrating PROVCREATOR's ability to generate more representative synthetic graphs that improve classification performance for underrepresented categories.

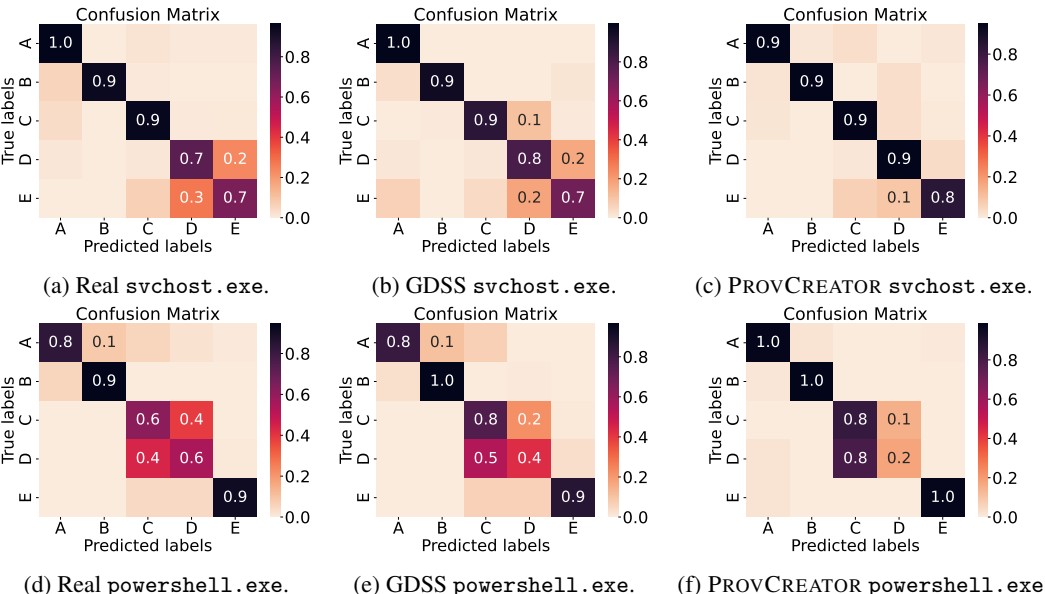

(a) Real `svchost.exe`.     (b) GDSS `svchost.exe`.     (c) PROVCREATOR `svchost.exe`.

(d) Real `powershell.exe`.     (e) GDSS `powershell.exe`.     (f) PROVCREATOR `powershell.exe`.

Figure 5: PROVCREATOR generated under-represented samples helped the ML model better classify `svchost.exe` and `powershell.exe` than GDSS generated samples.

Table 4: Precision, recall, F1 score, false positive rate (FPR), and area under receiver operating characteristic curve (ROC-AUC) for FLASH (Rehman et al., 2024) on an `svchost.exe` malware dataset (Mukherjee et al., 2023). Detection thresholds were chosen to maximize F1 score on the test set.

| Training Data | Precision ↑ | Recall ↑ | F1 Score ↑ | FPR ↓ | ROC-AUC ↑ |
|---|---|---|---|---|---|
| Real | 1.0 | 1.0 | 1.0 | 0.0 | 1.0 |
| GDSS + Real | 0.891 *(-0.109)* | 1.0 *(+0.000)* | 0.943 *(-0.057)* | 0.122 *(+0.122)* | 0.981 *(-0.019)* |
| **PROVCREATOR** + Real | **0.996** *(-0.004)* | **1.0** *(+0.000)* | **0.998** *(-0.002)* | **0.004** *(+0.004)* | **0.998** *(-0.002)* |

**Malware Detection.** Malware detection is an unsupervised learning task in which the model aims to identify program behaviors that are dissimilar to those represented in its training set. The malware samples in the test set were downloaded from VirusTotal and executed in a sandbox to collect relevant system provenance traces. Table 4 shows that PROVCREATOR's synthetic data induces less degradation in malware detection efficacy than that of prior works (Jo et al., 2022). The baseline in this case is incredibly strong — a perfect detector — because the malware behavior diverges significantly from `svchost.exe`'s typical behavior patterns, resulting in easily detectable differences in graph structure and node attributes. In future work, we will enrich this comparison with more challenging malware detection datasets.

## 5 CONCLUSION

In this paper, we introduced PROVCREATOR, a novel synthetic graph generation framework designed for heterogeneous graphs with nonuniform text attributes. By jointly considering the graph structure, node attributes, and program class labels, PROVCREATOR addresses program class imbalance in system provenance datasets with synthetic graphs that contain rich textual attributes. Further, the inclusion of flexible node attribute indicators enables the generation of semantically nuanced textual attributes, with multiple attributes per node. Our evaluation on real-world programs demonstrates improved graph structural fidelity compared to prior works, as well as improved utility in security-relevant downstream tasks. PROVCREATOR marks a step towards applying synthetic data generation to support accuracy-critical applications that require rich graph attributes.

**Reproducibility.**

All experimental code related to PROVCREATOR is available at https://anonymous.4open. science/r/provcreator-aio-4F83.

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

## A APPENDIX

### A.1 CHOSEN SUBPROGRAMS OF SVCHOST.EXE AND POWERSHELL.EXE

We chose svchost.exe and powershell.exe programs because they are popular targets for impersonation by advanced cyber threats (Barr-Smith et al., 2021), and their behavior is largely determined by their command-line arguments, which provides clear class labels.

Table 5: Label descriptions for svchost.exe.

| Label | Command | Description |
|---|---|---|
| A | svchost.exe -k netsvcs -p -s (ServiceName) | Starts a service in the netsvcs service group. This group contains services that interact with the network, including Windows Update (wuauserv), group policy client (gpsvc), and various other services. |
| B | svchost.exe -k LocalSystemNetworkRestricted -s (ServiceName) | Starts a service in the LocalSystemNetworkRestricted service group. This group contains services that do not connect to the network, like the File History Service (fhsvc) and the Portable Device Enumerator Service (WPDBusEnum). |
| C | svchost.exe -k wsappx -p -s (ServiceName) | Starts a service in the wsappx service group. This group contains the Client License Service (ClipSVC) and AppX Deployment Service (AppXSvc), which are both used in relation to AppX packaged applications installed from the Windows Store. |
| D | svchost.exe -k wusvcs -p -s WaaSMedicSvc | Starts the Windows Update Medic Service in the wusvcs group. This service automatically repairs components of Windows Update that are broken or disabled. |
| E | svchost.exe -k NetworkService -p -s DoSvc | Starts the Delivery Optimization service (DoSvc), which optimizes content delivery like updates. It runs under the NetworkService group. |

Table 6: Label descriptions for `powershell.exe`.

| Label | Command | Description |
|-------|---------|-------------|
| A | `powershell.exe -ExecutionPolicy AllSigned -NoProfile -NonInteractive -Command "& {Inline Script}"` | Executes an inline PowerShell script used by Windows Defender to verify the hash of scripts it uses. |
| B | `powershell.exe -ExecutionPolicy Bypass -NoProfile -Command "Add-Type 'C# Code';...` | Adds a .NET type, then calls a function from that type. These PowerShell commands are found in a Windows Defender script. |
| C | `powershell.exe -ExecutionPolicy Restricted -Command Write-Host 'Final result:  1';` | Runs a simple command that writes text to the console. |
| D | `powershell.exe -ExecutionPolicy Restricted -Command $Res = 0; $Infs = Get-Item ...` | Executes a PowerShell script that checks if an INF file with a certain key exists, and prints the result to console. |
| F | `powershell.exe /C Get-AppxPackage ...` | Lists installed AppX packages. |

## A.2 BLEU+

**BLEU+** is a metric specifically designed to measure the quality of filepath generation. Given a path $p$ splitted into segments of directories and filename $p_i \in p$ and a set of reference filepath $P^{\text{ref}}$, BLEU+ is defined as:

$$\text{bleup} = \max_{p_i^{\text{ref}} \in P^{\text{ref}}} \frac{\text{LCP}(p, p_i^{\text{ref}})}{|p_i^{\text{ref}}|}$$

where $|p|$ is the length (number of segments) of the filepath $p$, $\text{LCP}(\cdot, \cdot)$ is a function that calcualtes the segment level longest common prefix for two given string. Specifically, we define the equality of two segments in a way that better suits the scenario of filepath. The rules are:

- If the segment are identical, return true.

- If the segment is a filename (last segment) and the reference segment ends with ".`tmp`", return true.

- If the previous segment in the reference filepath is "`tmp`", "`temp`" or "`temporary`" (case insensitive), return `true`.

- If the previous segments in the reference filepath is "/home", "/mnt/home", "C: Users", return `true`.

- If both the reference segment and the input segment match the pattern of UUID, return true.

- Otherwise, return `false`.

## A.3 MODEL ARCHITECTURE

Figure 6 shows the architecture of the structure generation model. In our implementation, we set $L = 12$, $H = 8$, the dimension of $h$ is 256, the dimension of $e$ and $y$ is 128. The orange arrow highlights the conditional generation components we introduced.

Figure 7 shows the architecture of the structure generation model. In our implementation, we set $L = 4$, $M = 12$, hidden dimension of the GCN is 512, Embedding size of the transformer is 512 and max length is 128. We used the RobertaTokenizer which gives us the token size of 50265.

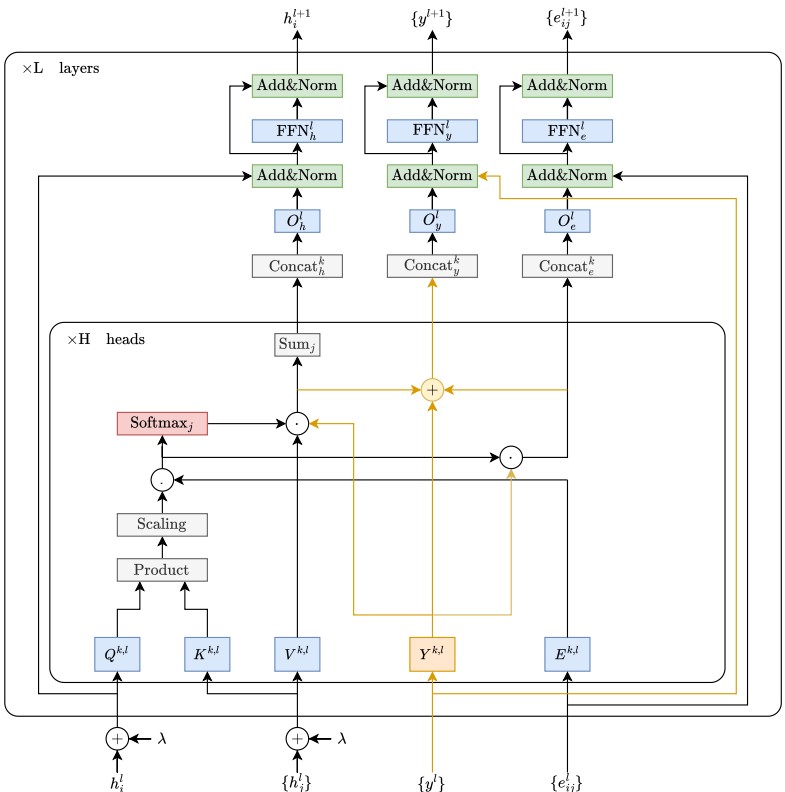

Figure 6: Structure Generation Model Architecture

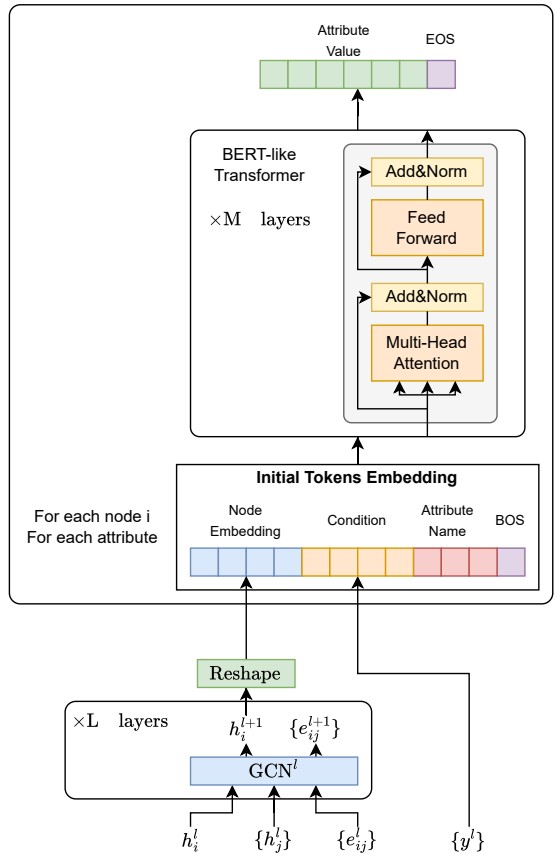

Figure 7: Attribute Generation Model Architecture

### A.4 EXAMPLE OF GENERATED GRAPHS

Figure 8 shows examples of generated graphs. The left column shows the generated structure, and the right column shows the generated attributes.

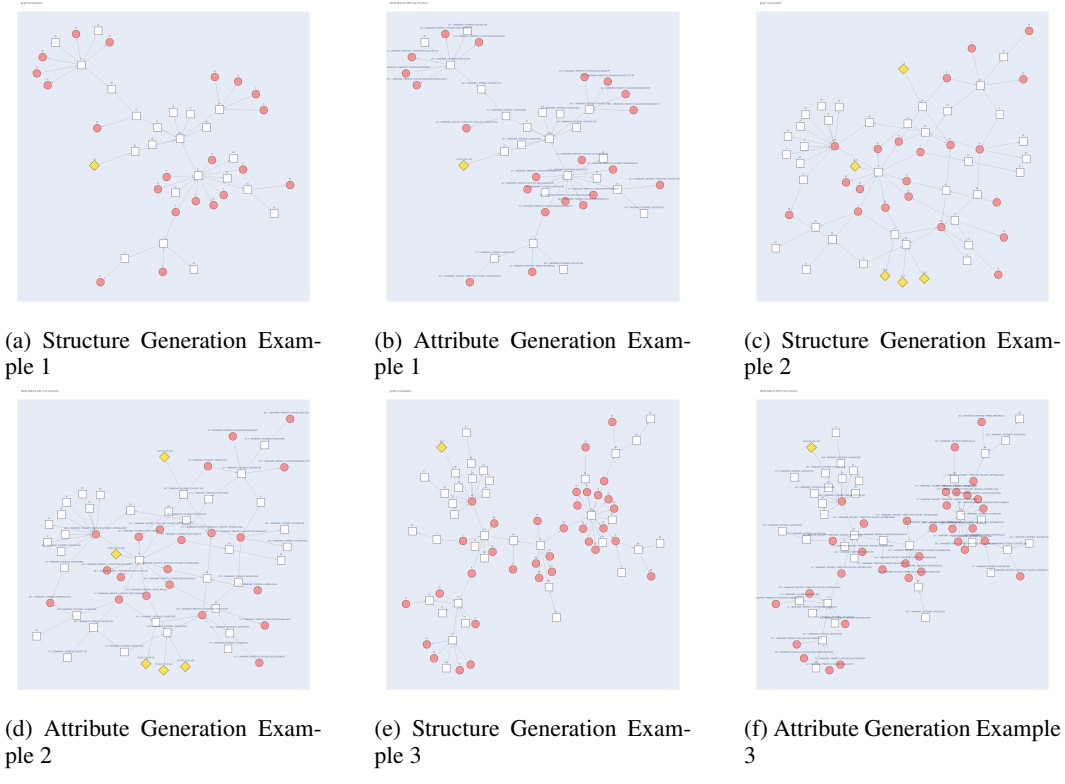

(a) Structure Generation Example 1

(b) Attribute Generation Example 1

(c) Structure Generation Example 2

(d) Attribute Generation Example 2

(e) Structure Generation Example 3

(f) Attribute Generation Example 3

Figure 8: Examples of generated graphs

