# OpenReview forum: "ProvCreator: Synthesizing Graph Data with Text Attributes"
_ICLR.cc/2025/Conference — Submitted to ICLR 2025_

### Official Review · Reviewer_eTGf · 2024-10-31

**Soundness:** 2
**Presentation:** 3
**Contribution:** 2
**Rating:** 5
**Confidence:** 4

**Summary:**

This paper addresses a fundamental data imbalance challenge in provenance graph construction -- different variations of syscalls are not equally represented in datasets, thus making it difficult for ML to model the nuances accurately. They propose ProvCreator to synthesize provenance graphs to oversample underrepresented versions of syscalls that still look similar enough to the original data.

**Strengths:**

This paper attempts to solve an important challenge for domains that suffer from imbalanced datasets. It is a one-shot approach to create both the graph structure and node attributes, which should arguably be cheaper than doing it individually (no supporting evidence though). The writing is clear.

**Weaknesses:**

The biggest weakness of this paper is the lack of correctness verification -- do the synthetic graphs even make sense? Are they realistic? Sure, checking whether graphs look structurally similar is one part of the evaluation but for the synthetic dataset to be useful in practice, it has to be semantically correct as well, which was never verified.

The malware detection downstream task is too shallow and vague to be useful to readers. Even the task "malware detection" is vaguely defined (see questions). Malware detection is an extremely challenging task, and for a baseline to achieve 100% F1-score almost always indicates that something has gone terribly wrong.

There is a fair amount of redundancy in the text (e.g., every time node attributes are mentioned, the same example set is used, ...). Once these are removed, there will be sufficient free space to provide examples of some of these synthetically created graphs.

**Questions:**

- One risk with oversampling works is that they create an inflated data distribution for the classes that naturally occur infrequently in practice. Having an unrealistic class distribution has a negative impact on generalization in open world settings. In this sense, why not embrace the imbalance and develop approaches that honour true class distributions?

- As previously mentioned, there are no experiments to show the semantic correctness of the generated graphs. Did the authors see cases where edges between nodes were created that make no sense, or some node attributes created that have no real-world significance? For instance, the port numbers could be classic example as they have semantic meaning beyond being a number. Operations valid on port 443 are not necessarily valid on port 8898. How does the attribute generation approach handle such semantic irregularities?

- In regards to malware detection, what is the exact classification task? Is it a binary class classification (malware vs svchost) or multiclass classification (malware families vs benign processes). What makes the classification that easy for a baseline to achieve 100% F1 score (as in real world, it is never this easy)?

- As far as I understand, the synthetic graphs are simply variants of graphs that already exist (no true generative approach). In this sense, how is this pipeline any better than, let's say, an evolutionary algorithmic approach? Are there benefits to a transformer-based approach that other popular "variant-creation" approaches lack?

- In Sec 3.1, the adjacency matrix is represented as real numbers. Why is that? What does that mean in terms of provenance graphs where a process opens a file. What does the real number represent here?

- This paper claims to develop a method purely for cybersecurity applications, yet I do not see anything in the approach that suggests its application purely to one domain. Could the authors provide clarification on this point?

---

> ### Author Response · Authors · 2024-11-23
>
> Below, we address the specific points raised and clarify our design choices. We are currently running extended evaluations.
>
> 1. Motivation for Balancing Classification Training Set: By balancing the training set distribution with synthetic data, we avoid overemphasizing the prior distribution of class labels in the training set. In our classification evaluation, we show that balancing the training distribution with synthetic data improves classification accuracy on a test set that follows the original distribution.
>
> 2. Semantic Correctness: We recognize both the necessity and difficulty of measuring the semantic validity of synthetic provenance graphs. While we cannot guarantee the semantic validity for the generated graphs, we can account for simple violations, such as semantically invalid edges (e.g. a file reading another file), which are discarded after generation. Similarly, obviously invalid generated attributes are also removed and replaced with a fallback value. To estimate the semantic validity of the generated node attributes, we developed metrics that incorporate domain-specific notions of correctness, accounting for varying program locations, random high-numbered ports, and random temporary file names. Despite our evaluation efforts, we acknowledge that we are unable to enumerate and measure all the possible ways a graph could become semantically invalid. Still, we observe that the inclusion of synthetic training data improves program classification F1 scores by 16%, indicating that the synthetic graphs are useful.
>
> 3. Malware Detection Task Clarification: The malware detection task is unsupervised binary classification between provenance graphs centered on benign and malicious SVCHOST.EXE processes. The training set consists entirely of benign graphs, and the test set consists of an equal number of benign and malicious graphs. We use FLASH (Rehman et al., 2024) as our detection system, which trains a small GCN model to perform node type classification, then compares the node type classification error rate against a threshold to determine if a graph is benign or malicious. We acknowledge that the baseline performance was too strong on the simple malware cases in our evaluation, and we are currently evaluating more diverse program and malware samples.
>
> 4. Synthetic Graph Generation: Building off of GDSS (Jo et al., 2022), ProvCreator generates synthetic graphs by iteratively diffusing from random noise. This approach is distinct from variant generation, where transformations are iteratively applied to a real seed graph. While both synthetic graph generation and variant-based graph augmentation are effective methods for improving the performance of graph learning models, we focus on graph synthesis in anticipation of long-term benefits of generative modeling research. As for the evolutionary algorithmic approach, deep learning based approaches have significant advantages in computational cost during the generation phase, and conditional generation improves control over the generated graphs.
>
> 5. Adjacency Matrix Clarification: The real number relaxation is necessary for GDSS because the diffusion step operates on the adjacency matrix, requiring it to be differentiable. There do exist other methods that use sparse diffusion steps and do not require the real number relaxation, such as [1]. Intuitively, a real number adjacency value can be interpreted as the probability of the existence of an edge. For example, for a process node $p$ and file node $f$, the entry $A[f, p, read] = 0.98$ indicates that $p$ has read $f$ with high probability.
>
> 6. Domain Generalization: We frame ProvCreator in the system provenance domain because that is our area of expertise. While our technique does not make domain-specific assumptions, we wish to avoid overclaiming domain generalization that is not explored in our evaluation. Future works may apply ProvCreator to other graph domains.
>
> [1] Sparse Training of Discrete Diffusion Models for Graph Generation (https://arxiv.org/abs/2311.02142)

---

> > ### Comment · Reviewer_eTGf · 2024-11-25
> >
> > I appreciate the response from the authors. I would advise to add these responses in the text to make it clearer for readers. I have updated my rating. Since the results from the malware domain are not present yet, I cannot score the paper any higher.

---

### Official Review · Reviewer_P6WW · 2024-11-03

**Soundness:** 3
**Presentation:** 2
**Contribution:** 3
**Rating:** 6
**Confidence:** 2

**Summary:**

This paper presents PROVCREATOR, a graph synthesis framework tailored for system provenance graphs, which addresses the critical issue of data imbalance in cybersecurity datasets by generating synthetic, balanced data that accurately reflects underrepresented program behaviors. By jointly learning the distribution of graph structures and node attributes conditioned on program class labels, ProvCreator not only mitigates systematic bias but also significantly enhances the performance of downstream ML models in classification and intrusion detection tasks. Furthermore, its flexible text attribute generation capability allows for the synthesis of diverse and contextually relevant attributes, making it a powerful tool for improving the reliability and effectiveness of security-critical applications that rely on rich graph attributes.

**Strengths:**

1. The paper addresses a critical and relevant topic.
2. The overall writing and structure are clear and coherent.
3. Experimental results demonstrate that the proposed method achieves promising effectiveness.
4. The authors provide code and data, which supports reproducibility.

**Weaknesses:**

1. While the paper thoroughly demonstrates the method's effectiveness, it lacks a discussion and experimental analysis of the method's efficiency and resource consumption during training. These aspects are essential for a comprehensive assessment of the method's practical utility. Additionally, I suggest that the authors include detailed parameter configurations used in the experiments to aid in replicability.
2. The paper heavily relies on domain knowledge in network security, which may make certain aspects challenging for general readers to understand. For instance, in Figure 2, the term “Given a graph class label” might benefit from further clarification. Could the authors provide specific examples illustrating the exact inputs and outputs at each stage?
3. Table 4 indicates that, in Malware classification, adding generated traceability graphs appears to degrade classification accuracy. Could the authors explain the reasons behind this performance drop and provide additional clarification?

**Questions:**

1. Why does the addition of provenance graphs lead to a decrease in Malware classification performance?
2. Could the authors provide an example to illustrate each step of the process described in Figure 2, including the exact form of the generated content at each stage?

---

> ### Author Response · Authors · 2024-11-23
>
> Below, we address the specific points raised and clarify our design choices. We are currently running extended evaluations.
>
> 1. Pipeline Scalability and Overhead: To address concerns about efficiency and resource consumption during training, we outline the computational overhead of our project pipeline as follows:
> - Data Pre-processing: Extracting ~120k training graphs takes ~48 hours, performance is limited by storage speed
> - Structural Model Training: ~48 hours on 4x 4090
> - Attribute Model Training: ~150 hours on 8x A6000
> - Generation: ~2 minutes per graph on 4x 4090
>
> 2. Below are the detailed input and output of each stage mentioned in Figure 2:
> - Graph label: The one-hot encoded labels used in our models are shown in Tables 5 and 6 in Appendix A.1.
> - Graph Structure Generator: The graph structure generator takes the POI process’s command with arguments as a condition prompt to generate a synthetic graph structure with only node type information. Examples of generated structure have been added to appendix section A.4.
>
> - Graph Encoder: The graph encoder takes a graph structure as input and creates a 512-wide embedding of for each node. These embeddings will be connected to the input to the transformer model, enabling joint training (Algorithm 1).
> The graph encoder is trained on real graph structures, but during inference it uses generated graph structures.
> - Transformer: For each node, the transformer model takes the node embedding, graph label, node type, and attribute name as input, and outputs an attribute value.  Examples of generated structure have been added to appendix section A.4.
>
> 3. Our malware detection experiments use an unsupervised anomaly detection approach. While increasing training data variation with synthetic data typically causes anomaly detection models to become more permissive, our baseline detector was too strong, leaving no room to reduce false positives. We are evaluating more challenging malware types to better showcase the value of synthetic provenance data.

---

### Official Review · Reviewer_61wY · 2024-11-03

**Soundness:** 2
**Presentation:** 3
**Contribution:** 2
**Rating:** 5
**Confidence:** 4

**Summary:**

This paper introduces ProvCreator, a graph synthesis framework specifically designed for feature-rich system provenance graphs within the cybersecurity domain. ProvCreator uniquely integrates the generation of graph structures with the prediction of node attributes by learning their joint distribution, thereby addressing the issue of underrepresented programs. The evaluation demonstrates that ProvCreator produces graphs with higher structural fidelity compared to the previous work it extends and enhances performance in subprogram classification tasks.

**Strengths:**

1. **Originality**: This work first considered together graph structure generation and attribute generation using joint training. The evaluation showed the approach benefited in some aspects.
2. **Significance**: This work researched a critical question.

**Weaknesses:**

1. **Unclear motivation**: The authors did not explain the reasonableness of combining graph structure generation and attribute generation together. I would liek to see further explanation rather than only mentioning that "neighborhood structures are often statistically codependent with node attributes."

2. **Unclear approach description**: The approach should be explained in more detail in section 3. As the authors claimed that they improved the backbone model and first used joint training in this field, the overall model architecture and the formula of joint loss function should be emphasized.

3. **Few baselines**: This work only chooses GDSS as its baseline. More previous works should be selected and compared, e.g., EDP-GNN[1], GraphEBM[2], etc. Besides, this work conducts experiments on a specific field and focuses on overcoming the data imbalance, so straightforward oversampling should also be considered as a baseline to show the meaningfulness of synthesis in this scenario.

4. **Few evaluated scenarios**: For graph similarity measures, only `svchost.exe` and `powershell.exe` in the Windows environment were evaluated. The generalization among different scenes and OSs was not proven.

5. **Questionable experiment design**: The overall performance of graph similarity was never compared with previous works because part of the baseline approach was replaced with random selection. The baseline would have more persuasiveness using previous works or self-constructed approaches from previous works, e.g., the evaluation in [3].

6. **No ablation experiment**: Ablation experiments correlated with novel aspects of this approach should be done, e.g., the adjustment of the backbone model and joint training.
----
1. Niu, Chenhao, et al. "Permutation invariant graph generation via score-based generative modeling." International Conference on Artificial Intelligence and Statistics. PMLR, 2020.
2. Liu, Meng, et al. "Graphebm: Molecular graph generation with energy-based models." arXiv preprint arXiv:2102.00546 (2021).
3. Chen, Xu, et al. "Node Attribute Generation on Graphs." arXiv preprint arXiv:1907.09708 (2019).

**Questions:**

1. For graph similarity measures, only `svchost.exe` and `powershell.exe` in the Windows environment were evaluated. Could you please provide some evidence of this work's generalization among different scenes and OSs?
2. It Seems like ProvCreator performed better than GDSS in the Malware Detection downstream task. Were there any correlations between the features of ProvCreator synthesized training data and newly solved test cases?

---

> ### Author Response · Authors · 2024-11-23
>
> Below, we address each of the points raised and provide clarifications regarding our design choices. We also outline ongoing and future efforts to enhance our evaluation and provide more comprehensive evidence.
>
> 1. Motivation for Combining Graph Structure and Attribute Generation: The core motivation of this work is to create realistic synthetic system provenance graphs. By combining graph structure with attribute generation, we capture dependencies between node attributes (e.g., program names) and graph structure (e.g., program actions). This way, we enhance the fidelity of the synthetic provenance graphs and improve their applicability to downstream detection tasks.
>
> 2. Insufficient Detail on Model Architecture and Joint Loss Function: The detailed model architecture has been added to appendix section A.3. Full source code is also available in anonymous repo https://anonymous.4open.science/r/provcreator-aio-4F83 and will be open sourced. The loss function for joint optimization is cross entropy for masked language modeling (sum of token level cross-entropy while ignoring masked tokens).
>
> 3. Evaluation of Baseline Graph Synthesis Techniques: We selected GDSS (Jo et al., 2022)  as our structural generation foundation because it has already been shown to beat existing structure-only approaches, such as EDP-GNN and GraphEBM. The core contribution of ProvCreator is the addition of structure-aware textual node attribute generation. The choice of graph structure generation method is orthogonal to our primary contribution. Further, constructing plausible synthetic text attribute baselines beyond sampling real node attributes is nontrivial compared to synthetic numeric or categorical attributes. In the table below, we find that ProvCreator’s synthetic graphs provide comparable performance gains to traditional class imbalance correction methods, demonstrating the effectiveness of class-conditioned targeted synthetic graph generation.
>
> | Training Data     | SVCHOST.EXE F1 Score | POWERSHELL.EXE F1 Score | /usr/bin/python3 F1 Score |
> | ----------------- | -------------------- | ----------------------- | ------------------------- |
> | Real Only         | 0.87                 | 0.72                    | 0.55                      |
> | Real + Synthetic  | 0.95                 | 0.84                    | 0.92                      |
> | Real, Oversampled | 1.00                 | 0.82                    | 0.91                      |
>
> 4. Generalization Across Operating Systems and Environments: While our current experiments focus on Windows processes, we are evaluating ProvCreator on provenance graphs from Linux environments and additional reference programs. Preliminary results suggest that ProvCreator easily adapts to OS-specific behaviors.
>
> 5. Correlation Between Synthesized Data and Downstream Results: ProvCreator outperforms GDSS due to its ability to incorporate text-based attributes conditioned on graph structure. We are extending our evaluation to include more diverse malware samples. We will also perform a case study of detection results improved by synthetic training data to clarify the influence of synthetic provenance graphs on the malware detection model.

---

> > ### Comment · Area_Chair_ULQz · 2024-11-26
> >
> > Dear Reviewer, the ICLR discussion period is extended. Could you please take a look at the authors' rebuttal and other reviews, and see whether you would like to update your ratings? The authors would greatly appreciate your consideration and responses.

---

> > > ### Comment · Reviewer_61wY · 2024-12-02
> > > **Feedback**
> > >
> > > Thank you for your rebuttal. It seems to me that the rebuttal does not directly address my questions. For example, for the motivation, there is still no supporting evidences for its argument. Similarly, the argument on OS-specific behaviors do not have any evidence to support. As such, I tend to keep my rating.

---

### Official Review · Reviewer_wDj6 · 2024-11-04

**Soundness:** 3
**Presentation:** 4
**Contribution:** 2
**Rating:** 5
**Confidence:** 4

**Summary:**

ProvCreator provides a framework designed to generate synthetic, feature-rich provenance graphs to address data imbalances in cybersecurity datasets. This is useful for intrusion detection and programming classification. Pre-existing datasets often lack representation of certain program behaviors. ProvCreator learns and generates graph structures and node attributes conditioned on program class labels, for synthetic graph generation using graph transformer-based diffusion approach for accurate synthesis.

ML models trained on data augmented with PROVCREATOR's graphs outperform those using older graph synthesis methods in program classification and malware detection tasks. PROVCREATOR’s synthetic graphs demonstrate higher structural and attribute fidelity, ultimately supporting better model generalization and performance in security-focused applications.

**Strengths:**

The approach of learning both the graph structure and node attributes conditioned on program class labels sets this work apart.
context-aware graph synthesis using diffusion is also a very interesting method. the flexibility to generate diverse text attributes tied to specific node types—addresses a gap in current provenance graph augmentation.

**Weaknesses:**

The need for text-based attributes conditioned on the graph's structure seems like a niche value-add, only relevant for training models that rely heavily on extremely specific node attributes to detect specific behaviors or patterns in cybersecurity applications. Your results of figure 5 and table 4 are promising in their extension of GDSS, but they are limited to two EXE files. This seems unusual - it might be more convincing to provide the results on a dataset of malware exe files, or something of that nature.

For instance, adaptive malware or threat actors often modify behavior in subtle ways that challenge even high-fidelity graph synthesis. While this framework builds on diffusion-based methods, these methods often fall short when modeling highly dynamic and adaptive attack behaviors that are common in real-world cybersecurity scenarios. Diffusion-based approaches can struggle with capturing temporal dependencies or the evolving nature of interactions within system provenance graphs, which can limit their applicability to detecting sophisticated, adaptive threats. That’s why other work has turned to methods that incorporate temporal modeling or hybrid architectures combining diffusion processes with recurrent or attention-based networks, which could provide more flexibility for handling complex, evolving data structures.

It's always worth addressing the limitations of static analysis, which can miss context-sensitive behavior or runtime characteristics that only emerge during execution.

Scalability is another area that needs more attention. While PROVCREATOR shows results in generating complex, high-fidelity graphs, the paper doesn’t dive into the potential computational costs involved, especially as datasets scale up. Generating graphs with detailed structure and attributes can become resource-intensive, raising questions about runtime efficiency and memory usage in larger, real-world applications. A clearer analysis of how PROVCREATOR handles these demands, or what optimizations could be implemented to make it more scalable, would go a long way in proving its viability for real-world cybersecurity operations where resources are always a factor.

**Questions:**

Do you think integrating some form of temporal modeling could help address the limitations in capturing dynamic behavior?

"While our work only considers static attribute sets for each node type, PROVCREATOR’s architecture naturally allows for dynamically determined node attribute sets using flexible attribute indicators." - could you elaborate? did you do any experimentation for this?

"The original GDSS paper (Jo et al., 2022) uses a GCN as the backbone and it does not consider conditional generation. We improved the backbone model by using a graph transformer (Dwivedi & Bresson, 2020) and following the original stable diffusion (Rombach et al., 2021) paper’s approach to add the conditioning mechanism for the graph transformer. The condition is set based on the label of the process represented in the graph; details about the labels can be found in §A.1." could you please emphasize how your work significantly differs from prior work, especially from GDSS?

---

> ### Author Response · Authors · 2024-11-23
>
> Below, we address the specific points raised and clarify our design choices. We are currently running extended evaluations.
>
> 1. Text-based Attributes Conditioned on Graph Structure: Node attributes are related to the graph’s structure in many domains. For instance, in biology, text-based attributes describing molecular interactions or protein functions can be generated from molecular interaction networks. Similarly, in knowledge graphs, user-item node attributes significantly influence recommendation decisions.
>
> 2. Pipeline Scalability and Overhead: The computational overhead of the project pipeline is structured as follows:
>  - Data Pre-processing: Extracting ~120k training graphs takes ~48 hours, performance is limited by storage speed
>  - Structural Model Training: ~48 hours on 4x 4090
>  - Attribute Model Training: ~150 hours on 8x A6000
>  - Generation: ~2 minutes per graph on 4x 4090
>
> 3. Dynamic Behavior and Temporal Modeling: Provenance graphs capture dynamic runtime behaviors by summarizing low-level system logs, using directed edges to represent information flows between system resources. Temporal modeling may improve the alignment between the graph synthesis framework and the process of capturing real provenance graphs, representing an interesting direction for future work.
>
> 4. Dynamic Node Attributes in ProvCreator: Because our evaluation focuses on the system provenance domain, where node attribute sets are determined by the node type, we do not have a practical downstream application for dynamically determined node attribute sets. While ProvCreator’s design is domain-agnostic and dynamic node attribute sets may be important in some domains, this capability is not a core contribution of ProvCreator. Evaluating synthetic graphs with dynamically determined node attribute sets may be explored in future works.
>
> 5. Comparison with GDSS: Our work improves GDSS in two key aspects:
> *(1)* Backbone Improvements: We leverage a powerful GNN model, GraphTransformer, to improve the representation and generation process; *(2)* Conditional Generation: We added conditional generation capability so we can specify the type of the graph we want to generate. The types are defined based on the command line of the POI process, detailed in Appendix A.1.
> These enhancements allow us to address limitations in GDSS while achieving more robust conditioning and attribute synthesis.

---

> > ### Comment · Area_Chair_ULQz · 2024-11-26
> >
> > Dear Reviewer, the ICLR discussion period is extended. Could you please take a look at the authors' rebuttal and other reviews, and see whether you would like to update your ratings? The authors would greatly appreciate your consideration and responses.

---

### Meta-Review · Area_Chair_ULQz · 2024-12-21

**Metareview:**

This submission introduces ProvCreator, a graph synthesis framework for applications in cybersecurity. The motivation is that existing provenance datasets have limitations in terms of imbalanced program representation. The proposed method aims to learn the joint distribution of node attributes and graph structures based on program class labels, thus allowing for the generation of realistic provenance graphs to balance underrepresented programs. The authors' evaluation shows that the proposed method creates graphs with better structural and attribute fidelity, as well as improved downstream utility.

The reviewers identified the positive parts of this work as:
- the idea of joint graph structure and attribute generation is useful (Reviewers wDj6, 61wY, eTGf).
- the studied problem aims to address an important challenge in cybersecurity related to data imbalance and graph synthesis (Reviewers 61wY, eTGf).
- the paper is relatively well-written, and code and data are provided for reproducibility (Reviewers P6WW, eTGf).

The weaknesses raised include the following:
- the paper lacks sufficient baseline comparisons, evaluated scenarios, and ablation studies (Reviewers 61wY, P6WW).
- the motivation for combining graph structure and attribute generation is not fully explained (Reviewers 61wY, P6WW).
- lack of analysis of resource efficiency, scalability, or the semantic correctness of generated graphs (Reviewers wDj6, P6WW, eTGf).

During the rebuttal stage, the overall engagement is not particularly enthusiastic.
- reviewer 61wY (5) considered that the rebuttal does not directly address their questions. "For example, for the motivation, there is still no supporting evidences for its argument. Similarly, the argument on OS-specific behaviors do not have any evidence to support. As such, I tend to keep my rating."
- reviewer eTGf (5) updated rating from 3 to 5. "Since the results from the malware domain are not present yet, I cannot score the paper any higher."
- it is unfortunate that reviewers wDj6 (5) and P6WW (6) did not responded to the authors' rebuttal.

In summary, this work has final ratings of 6, 5, 5, 5. The reviewers' main concerns on empirical demonstration and technical motivation / presentation remained unaddressed after the rebuttal. Based on this reason, this work in its current form is not recommended for acceptance, since it would significantly benefit from another round of revision.

**Additional Comments On Reviewer Discussion:**

During the rebuttal stage, the overall engagement is not particularly enthusiastic.
- reviewer 61wY (5) considered that the rebuttal does not directly address their questions. "For example, for the motivation, there is still no supporting evidences for its argument. Similarly, the argument on OS-specific behaviors do not have any evidence to support. As such, I tend to keep my rating."
- reviewer eTGf (5) updated rating from 3 to 5. "Since the results from the malware domain are not present yet, I cannot score the paper any higher."
- it is unfortunate that reviewers wDj6 (5) and P6WW (6) did not responded to the authors' rebuttal.

---

### Decision · Program_Chairs · 2025-01-22

Reject